# *Escherichia coli* ST131 clones harbouring AggR and AAF/V fimbriae causing bacteremia in Mozambican children: Emergence of new variant of *fimH27* subclone

Inácio Mandomando[1,2]*, Delfino Vubil[1], Nadia Boisen[3], Llorenç Quintó[4], Joaquim Ruiz[4,5], Betuel Sigaúque[1,2], Tacilta Nhampossa[1,2], Marcelino Garrine[1], Sergio Massora[1], Pedro Aide[1,2], Ariel Nhacolo[1], Maria J. Pons[4,6], Quique Bassat[1,4,7,8,9], Jordi Vila[4,10], Eusébio Macete[1,11], Flemming Scheutz[3], Myron M. Levine[12], Fernando Ruiz-Perez[13], James P. Nataro[13], Pedro L. Alonso[1,4]

1 Centro de Investigação em Saúde da Manhiça (CISM), Maputo, Mozambique, 2 Instituto Nacional de Saúde (INS), Ministério da Saúde, Maputo, Mozambique, 3 Department of Bacteria, Parasites and Fungi, Statens Serum Institut, Denmark, 4 ISGlobal, Hospital Clínic—Universitat de Barcelona, Barcelona, Spain, 5 Universidad Continental, Lima, Peru, 6 Laboratorio de Microbiología Molecular y Genómica Bacteriana, Universidad Científica del Sur, Lima, Peru, 7 ICREA, Pg. Lluís Companys 23, Barcelona, Spain, 8 Pediatric Infectious Diseases Unit, Pediatrics Department, Hospital Sant Joan de Déu, University of Barcelona, Barcelona, Spain, 9 Consorcio de Investigación Biomédica en Red de Epidemiología y Salud Pública (CIBERESP), Madrid, Spain, 10 Servei de Microbiologia, Centre de Diagnòstic Biomèdic, Hospital Clínic, Barcelona, Spain, 11 Direcção Nacional de Saúde Pública, Ministério de Saúde, Maputo, Mozambique, 12 Center for Vaccine Development, University of Maryland School of Medicine, Baltimore, Maryland, United States of America, 13 Department of Pediatrics, University of Virginia School of Medicine, Charlottesville, Virginia, United States of America

* inacio.mandomando@manhica.net, inacio.mandomando@gmail.com

**Data Availability Statement:** All relevant data are within the manuscript and its Supporting Information files.

## Abstract

Multidrug-resistant *Escherichia coli* ST131 *fimH30* responsible for extra-intestinal pathogenic (ExPEC) infections is globally distributed. However, the occurrence of a subclone *fimH27* of ST131 harboring both ExPEC and enteroaggregative *E. coli* (EAEC) related genes and belonging to commonly reported O25:H4 and other serotypes causing bacteremia in African children remain unknown. We characterized 325 *E. coli* isolates causing bacteremia in Mozambican children between 2001 and 2014 by conventional multiplex polymerase chain reaction and whole genome sequencing. Incidence rate of EAEC bacteremia was calculated among cases from the demographic surveillance study area. Approximately 17.5% (57/325) of isolates were EAEC, yielding an incidence rate of 45.3 episodes/$10^5$ children-years-at-risk among infants; and 44 of isolates were sequenced. 72.7% (32/44) of sequenced strains contained simultaneously genes associated with ExPEC (*iutA*, *fyuA* and *traT*); 88.6% (39/44) harbored the aggregative adherence fimbriae type V variant (AAF/V). Sequence type ST-131 accounted for 84.1% (37/44), predominantly belonging to serotype O25:H4 (59% of the 37); 95.6% (35/44) harbored *fimH27*. Approximately 15% (6/41) of the children died, and five of the six yielded ST131 strains (83.3%) mostly (60%; 3/5) due to serotypes other than O25:H4. We report the emergence of a new subclone of ST-131 *E. coli* strains belonging to O25:H4 and other serotypes harboring both ExPEC and EAEC

**Funding:** CISM receives core funding from "Agencia Española de Cooperacion Internacional para el Desarollo (AECID)". Joaquim Ruiz had a fellowship from the program I3SNS, of the ISCIII (grant number: CES11/012). Jordi Vila is supported by the Spanish Network for the Research in Infectious Diseases (REIPI RD16/0010). Nadia Boisen was funded by the Independent Research Fund Denmark (grant number: DFF1333-00156). The funders had no role in the study design, data collection and analysis, decision to publish, or preparation of the manuscript.

**Competing interests:** The authors declare that they have no conflict of interest.

virulence genes, including *agg5A*, associated with poor outcome in bacteremic Mozambican children, suggesting the need for prompt recognition for appropriate management.

## Author summary

*Escherichia coli* ST131 has emerged as a globally disseminated multi-drug resistant clone associated with extra-intestinal infections acquired in the community or hospital. In Manhiça district, *E. coli* is among the top five leading bloodstream pathogens in children. We characterized *E. coli* strains causing bacteremia in young children in a rural hospital of Mozambique, providing novel information on the occurrence of a new subclone of ST131 harboring both ExPEC and EAEC related genes and belonging to commonly reported O25:H4 and other serotypes. These data suggest the need for further understanding of pathogenesis and clinical impact of this new entity to inform prompt recognition and appropriate treatment.

## Introduction

*Escherichia coli* is a common cause of community and hospital-acquired bacterial infection, causing a wide range of clinical diseases and associated with high morbidity and mortality worldwide [1,2]. Two major groups of pathogenic *E. coli*—diarrheagenic *E. coli* (DEC) and extra-intestinal pathogenic *E. coli* (ExPEC)—are recognized, differing in their virulence factors and associated clinical syndromes [2]. ExPEC strains are among the major causes of urinary tract infections (UTI) and/or hospital/community-acquired bacteremia in both industrialized [3,4] and low-income countries [4–6]. These strains differ from DEC or commensal *E. coli* strains with respect to their virulence factors, with the former requiring specific attributes to cause invasive disease, e.g. the ability to survive in serum, efficient iron uptake mechanisms and internalization by the host [7].

In contrast, six well recognized pathotypes of DEC (enteropathogenic *E. coli* [EPEC], Shiga toxin (Stx)-producing *E. coli* [STEC], enterotoxigenic *E. coli* [ETEC], enteroinvasive *E. coli* [EIEC], enteroaggregative *E. coli* [EAEC] and diffusely adherent *E. coli* [DAEC]) [2], are among the leading etiological agents of childhood [8,9] and travelers' [10] diarrhea. Among the ExPEC, a wide range of pathogenic lineages of *E. coli* have been reported, including sequence type (ST) 131, among others [11,12]. ST131 firstly reported in the early 2000s, is a globally disseminated multidrug resistant (MDR) clone with serious clinical impact [13–15], particularly in Africa [12,16]. The most prevalent subclones of ST131 include *fimH*30 [14], belong to serotypes O25:H4 and O16:H4 [13] is widely reported including clonal diversity [16]. However, to our knowledge, the occurrence of ST131 subclone *fimH27*, exhibiting serotypes different from O25:H4 or O16:H4 in addition to harboring EAEC genes (e.g. aggR and AAF/V), remains obscure.

EAEC has long been regarded as an intestinal pathogen and therefore unlikely to cause disease in normal patients outside of the intestinal tract [17]. However, recent reports have linked EAEC strains with urinary tract infection [18] and fatal hemolytic uremic syndrome [19] outbreaks. The role of EAEC in extra-intestinal infections and associated outcomes among African children admitted to hospitals, remains unknown. Through our ongoing invasive bacterial disease surveillance, we previously documented *E. coli* as among the top five pathogens associated with community-acquired bacteremia in Mozambican children, with an associated case

fatality ratio of nearly 10% [6]. Thus, assessing the molecular virulence markers of *E. coli* strains circulating is an important first step in understanding the molecular epidemiology of this entity in Mozambique, with the hope of informing appropriate control or prevention strategies. Herein, we aim to characterize the molecular epidemiology of *E. coli* causing childhood bacteremia in a rural Hospital in Mozambique between 2001 and 2014.

## Materials and methods

### Ethics statement

Clinical data were routinely collected from an ongoing morbidity surveillance system in Manhiça district health facilities, established as part of CISM's HDSS and approved by the Mozambican Ministry of Health. All residents of the district of Manhiça have signed an individual informed consent to become part of the ongoing HDSS established in the area.

### Study population

The study was conducted by the "*Centro de Investigação em Saúde de Manhiça* (CISM)" at the Manhiça District Hospital (MDH), the main referral health facility for the Manhiça district, a rural area located 80 km north of Maputo, southern Mozambique. The district has an estimated population of 183,000 inhabitants, and in this area, CISM has been running a continuous health and demographic surveillance system (HDSS) since 1996, currently covering the entire district's population. A full description of the geographical and socio-demographic characteristics of the study community has been presented elsewhere [20]. Of importance, HIV sero-prevalence in the area is among the highest in the world (40% of the general adult population) [21]. CISM is adjacent to the MDH, and since 1997, the hospital and CISM have jointly operated a 24-hour surveillance of all pediatric (<15 years of age) visits to the outpatient department and admissions to the wards including surveillance of invasive bacterial disease as described previously [6].

### Sample collection and laboratory procedures

As part of routine clinical practice at MDH, a single venous blood specimen for bacterial culture was systematically collected upon hospital admission for all children <2 years of age, and for children aged 2 to <15 years with axillary temperature ≥39.0°C or with signs of severe illness as judged by the admitting clinician for bacterial isolation as detailed described elsewhere [6].

### Detection of diarrheagenic *E. coli* pathotypes, phylogenetic groups and virulence factors

Three hundred and twenty-five frozen *E. coli* isolates recovered from blood cultures were retrieved, sub-cultured on MacConkey, and screened for the presence of EAEC, ETEC, EPEC, and STEC markers by multiplex polymerase chain reaction (PCR) [22] including phylogenetic group as described elsewhere [23]. Additionally, we tested the isolates for various ExPEC and DEC virulence genes by conventional multiplex PCRs, targeting 44 genes including those commonly prevalent in EAEC (such as *aggR*, *aatA*, *aap*, *aaiC* and the recently discovered *aar* gene) [24,25]. Positive samples were confirmed by sequencing five isolates for each gene of interest in the Sequencing Core Facility of the University Of Virginia School Of Medicine, Charlottesville, VA, USA.

## Antimicrobial resistance and mechanisms of resistance

The antimicrobial susceptibility phenotype for ampicillin, amoxicillin-clavulanic acid, cefuroxime, ceftriaxone, cefotaxime, aztreonam, ertapenem, imipenem, meropenem, nalidixic acid, ciprofloxacin, chloramphenicol, amikacin, tobramycin, Gentamicin, Tetracycline and trimethoprim-sulfamethoxazole (SxT) was determined by a conventional disk diffusion method on Mueller Hinton agar [26] using commercially available disk (Oxoid, Basingstoke, Hampshire, UK). The interpretative category of resistance was determined according to the Clinical Laboratory Standard Institute (CLSI) guidelines [27]. Multidrug resistance was defined as resistance to three or more unrelated antibiotic families and we considered non-susceptible isolates those with an intermediate or full resistant profile [28]

## Serotyping and whole genome sequence (WGS)

Somatic (O) and flagella (H) antigens were phenotypically identified using commercially available antisera as described elsewhere [29,30]. The following designations were included: "O rough," the boiled culture auto-agglutinated, suggesting absence of O antigen; "O?" when it could not be determined whether the strain produces an O antigen (precipitation with Cetavlon indicates an acidic polysaccharide that could represent capsular K antigen); and "ONT," when the O antigen was found to be present but could not be typed. Serotyping was performed on all bloodstream isolates positive for EAEC markers and a subset of other *E. coli* bacteremic strains at the International *Escherichia* and *Klebsiella* Centre, Department of Bacteria, Parasites and Fungi, Statens Serum Institut (SSI), Copenhagen, Denmark. Additionally, serotypes were also assessed by WGS and compared to conventional serotyping.

Forty-four EAEC isolates (the total of positive by PCR) and 22 non-EAEC isolates for control purpose (randomly selected) were sequenced by using Illumina Miseq (Illumina, San Diego, CA, USA). Briefly, Genomic DNA from isolates was purified using Qiagen DNeasy Blood and Tissue kit (Qiagen, Valencia, USA) according to the kit protocol. Initial DNA concentrations were measured and quantified using Qubit Flourometer and dsDNA BR/HR Assay Kit (Thermo Fisher Scientific). Sample and library preparation were performed using the Nextera XT v2 DNA library Preparation Kit (Illumina, Sand Diego, USA). Libraries were finally purified by Agencourt AMPpure XP System (Beckman Coulter, Indianapolis, USA). WGS data were pre-processed employing a QC-pipeline (available at https://github.com/ssi-dk/SerumQC), where isolate sequences were removed in case of contamination with more than 5% of another genera, as well as sequences representing EAEC isolates with genome sizes outside the range of 4.64 Mbp-5.56 Mbp. Isolate sequences were removed from the dataset if assemblies comprised of more than 350 contigs. *De novo* assemblies were carried out using CLC Genomics Workbench 10 with a minimum contig length of 200 bp. The genome size, N50, and contigs are presented in S1 Table.

Sequence type, *in silico* serotype and virulence genes were determined from the *de novo* assembled genomes using the webtools available at https://cge.cbs.dtu.dk/services/ [31]. The least ambiguous phenotypical or *in silico* serotype was used in the final analysis *i.e.* non-motile strains (H-) were given the *in silico* determined *fliC* H type, the *in silico* O type was used on O rough and O? and the phenotypic O group was used if the *in silico* O type was ambiguous or non-typeable.

## Definitions

Bacteremia was defined as the isolation of at least one non-contaminant bacteria from the blood culture collected on admission. Bacteremic EAEC strains were defined as isolates from blood culture testing positive for one of the following genes: *aggR*, *aaiC* or *aatA* genes by

multiplex PCR. Other bacteremic *E. coli* were defined as *E. coli* strains from blood culture excluding EAEC, EPEC, ETEC and STEC.

Case fatality ratios (CFR), represent in-hospital mortality due to bacteremia calculated for admitted children with known outcome (i.e. discharged alive, or dead), excluding patients that left the hospital without medical permission or were transferred to Maputo Central Hospital as previously described [6].

## Statistical analysis

Statistical analyses were performed using STATA software, version 13.0 (Stata Corp., College Station, TX, USA). The proportion of virulence factors found among children infected with EAEC and other bacteremic *E. coli* was compared using Chi-squared or Fisher's exact tests as appropriate. Minimum community-based incidence rates (MCBIRs) for *E. coli* bacteremia (EAEC and other bacteremic *E. coli* excluding ETEC and EPEC) were calculated referring cases to population denominators establishing time at risk (child years at risk [CYAR]) inferred from the HDSS census information. Children did not contribute to the numerator or denominator for a period of 15 days after each episode or when they were outside the study area.

CART analysis was also performed, as previously described [24] (CART Pro Version 7.0; Salford Systems). We included the collective number of virulence genes present (virulence factor score, VFS) in putting 48 factors of interest as binary (present/absent) independent predictive variables along with a continuous "factor total" that was a sum of all factors including the presence of malaria. Alive/death was the binary dependent outcome variable for isolates causing bacteremia. Furthermore, the CART analyses for bacteremic EAEC were also assayed for 66 genes assessed by WGS.

## Results

### Screening for EAEC markers, serotypes and phylogenetic groups distribution

From January 2001 to December 2014, 37,536 blood cultures were collected from children younger than 3 years of age admitted to the MDH; and 325 (0.8%) were positive for *E. coli*. Of these, 57 (17.5%) met the definition of EAEC, 6 (1.8%) ETEC and 2 (0.6%) EPEC; while the remaining 260 (80.0%) were classified as other bacteremic *E. coli*. Children with EAEC bacteremia appeared to be younger than those with bacteremia secondary to other *E. coli* [mean: 10.1 months (SD = 7.2) *vs*. 12.4 (SD = 8.3), p = 0.057], albeit not statistically significance.

Sixty-six (20.3%) [44 EAEC and 22 other bacteremic *E. coli*] of the 325 *E. coli* isolates were serotyped, with serotype O25:H4 being the most frequent, accounting for 22 isolates (33.3%), followed by O127:H4 with six (9.1%) isolates, and O51:H4 and O86:H4 with four (6.1%) isolates each. All O25:H4, five of six O127:H4, and all O51:H4 and all O86:H4 were EAEC bacteremic *E. coli* isolates, followed by nine isolates (13.6%) of seven different serotypes (Table 1).

### Virulence factors detected by conventional multiplex PCR

Of the 57 EAEC, detailed virulence factors were assessed in the 44 isolates serotyped and sequenced. Of the 44 EAEC strains serotyped from blood, 41 (93.2%) met the definition of typical EAEC (presence of the master regulator gene, *aggR*) and were positive for *aar* (AggR- activated regulator). Only one typical EAEC (*aggR*+) strain lacked *aatA* while 37 showed the combined presence of virulence genes *aggR*, *aap*, *aatA* and ORF3; and five and two isolates lacked the *aap* or ORF3 genes, respectively. Table 2 shows the prevalence of virulence factors

**Table 1. Genomic characteristics of the 44 enteroaggregative *E. coli* strains isolated from children admitted at the MDH with bacteremia, analyzed by WGS.**

| C-number | Phenotypical serotype | Molecular serotype | ST | Fimtype | Virulence genes | AAF | Phylogeny | Age* | Outcome | Malaria | Diarrhea | Sex |
|---|---|---|---|---|---|---|---|---|---|---|---|---|
| C169-15 | O125ab/O176:H- | O176:H33• | 10 | H54 | *aaiC,astA,capU,gad,mchB,mchC,mchF, mcmA,ORF3,ORF4,pic,sat, fyuA, hra, iutA, agn43* | Neg. | A | 10 | Alive | UNK | Yes | F |
| C194-15 | O125ab/O176:H- | O176:H33• | 10 | H54 | *aaiC,astA,capU,gad,mchB,mchC,mchF, mcmA,ORF3,ORF4,pic,sat,fyuA, hlyA, hra, iutA, agn43* | Neg. | A | 12 | Alive | Yes | No | F |
| C190-15 | O11:H- | O11:H18• | 31 | Neg. | *aap,aar,aatA,aggA,aggB,aggC,aggD, aggR,air,astA,eilA,gad,iha,iss,lpfA,ORF3, ORF4,sat, fyuA, traT, papC, papA, hlyA, hra, papGII, iutA, agn43* | I | B2 | 9 | Alive | Yes | No | F |
| C188-15 | O153:H30 | H30• | 38 | H5 | *aap,aar,aggA,aggB,aggC,aggD,aggR, capU,eilA,gad,iss,nfaE,ORF3,ORF4, fyuA, traT, hra, agn43* | I | D | 3 | Alive | Yes | Yes | F |
| C175-15 | O44:H34 | O17/O77:H34• | 130 | H47 | *aap,aatA,agg4A,agg4C,agg4D,aggR,air, eilA,gad,lpfA,ORF3,ORF4,sepA, agn43* | IV | D | 9 | Alive | Yes | No | M |
| C159-15 | O25:H4 | O25:H4 | 131 | H27 | *aap,aar,aatA,agg3B,agg3C,agg3D,agg5A, aggR,cnf1,gad,iss,ORF3,ORF4, fyuA, papA, hlyA, hlyC, papGII, iutA, agn43* | V | B2 | 16 | Alive | No | No | M |
| C164-15 | O25:H4 | O25:H4 | 131 | H27 | *aap,aar,aatA,agg3B,agg3C,agg3D,agg5A, cnf1,gad,iss,ORF3,ORF4, fyuA, traT, papC, papA, hlyA, hlyC, aer, papGII, iutA, agn43* | V | B2 | 13 | Alive | No | No | M |
| C166-15 | Orough:H4 | O127:H4• | 131 | H27 | *aap,aar,aatA,agg3B,agg3C,agg3D,agg5A, aggR,cnf1,gad,iss,ORF3,ORF4, fyuA, traT, papC, papA, hlyA, hlyC, aer, papGII, iutA, agn43* | V | B2 | 30 | Alive | Yes | No | M |
| C167-15 | O25:H4 | O25:H4 | 131 | H27 | *aap,aar,aatA,agg3B,agg3C,agg3D,agg5A, aggR,cnf1,gad,iss,ORF3,ORF4, fyuA, traT, papC, papA, hlyA, hlyC, aer, papGII, iutA, agn43* | V | B2 | 28 | Alive | No | No | M |
| C168-15 | O?:H4 | O51:H4• | 131 | H27 | *aap,aar,aatA,agg3B,agg3C,agg3D,agg5A, aggR,cnf1,gad,iss,ORF3,ORF4, fyuA, traT, papC, papA, hlyA, papGII, iutA, agn43* | V | B2 | 9 | Alive | No | No | F |
| C170-15 | O25:H4 | O25:H4 | 131 | H27 | *aap,aar,aatA,agg3B,agg3C,agg3D,agg5A, aggR,cnf1,gad,iss,ORF3,ORF4, fyuA, traT, papC, papA, hlyC, papGII, iutA, agn43* | V | B2 | 4 | Dead | No | Yes | M |
| C171-15 | O25:H4 | O25:H4 | 131 | H27 | *aap,aar,aatA,agg3B,agg3C,agg3D,agg5A, aggR,gad,iha,iss,nfaE,ORF4,sat, fyuA, traT, papC, papA, hlyA, hlyC, aer, papGII, iutA, agn43* | V | B2 | 9 | Alive | Yes | Yes | M |
| C174-15 | O86:H4 | O86:H4 | 131 | H27 | *aap,aar,aatA,agg3B,agg3C,agg3D,agg5A, aggR,cnf1,gad,iss,ORF3,ORF4, fyuA, traT, papC, papA, hlyA, hlyC, aer, papGII, iutA* | V | B2 | 3 | Alive | Yes | No | M |
| C177-15 | O25:H4 | O25:H4 | 131 | H27 | *aap,aar,aatA,agg3B,agg3C,agg3D,agg5A, aggR,cnf1,gad,iss,ORF4, fyuA, traT, papC, papA, hlyA, hlyC, aer, papGII, iutA, agn43* | V | B2 | 8 | Alive | No | No | M |
| C178-15 | O25:H4 | O25:H4 | 131 | H27 | *aap,aar,aatA,agg3B,agg3C,agg3D,agg5A, aggR,cnf1,gad,iss,ORF3,ORF4, fyuA, traT, papC, papA, hlyC, aer, papGII, iutA, agn43* | V | B2 | 9 | Alive | No | Yes | M |

(*Continued*)

**Table 1.** (Continued)

| C-number | Phenotypical serotype | Molecular serotype | ST | Fimtype | Virulence genes | AAF | Phylogeny | Age* | Outcome | Malaria | Diarrhea | Sex |
|---|---|---|---|---|---|---|---|---|---|---|---|---|
| C179-15 | O127:H4 | O127:H4 | 131 | H27 | *aap,aar,aatA,agg3B,agg3C,agg3D,agg5A, aggR,cnf1,iss,ORF3,ORF4, fyuA, papA, hlyA, hlyC, aer, papGII, iutA* | V | B2 | 7 | Dead | No | No | M |
| C182-15 | O15:H4 | O15:H4 | 131 | H27 | *aap,aar,aatA,agg3B,agg3C,agg3D,agg5A, aggR,gad,iss,ORF3,ORF4, fyuA, traT, papC, papA, hlyA, aer, papGII, iutA, agn43* | V | B2 | 19 | Alive | No | No | M |
| C184-15 | O25:H4 | O25:H4 | 131 | H27 | *aap,aar,aatA,agg3B,agg3C,agg3D,agg5A, aggR,cnf1,gad,iss,ORF3,ORF4, fyuA, traT, papC, papA, hlyA, hlyC, papGII, iutA, agn43* | V | B2 | 7 | Alive | No | No | F |
| C185-15 | O25:H4 | O25:H4 | 131 | H27 | *aap,aar,aatA,agg3B,agg3C,agg3D,agg5A, aggR,cnf1,gad,iss,ORF4,pic, fyuA, traT, papC, papA, hlyA, hlyC, aer, papGII, iutA, agn43* | V | B2 | 8 | Transf. | Yes | No | M |
| C189-15 | O86:H4 | O86:H4 | 131 | H27 | *aap,aar,aatA,agg3B,agg3C,agg3D,agg5A, aggR,cnf1,gad,iss,ORF3,ORF4, fyuA, traT, papC, papA, hlyA, hlyC, aer, papGII, iutA, agn43* | V | B2 | 32 | Dead | No | No | M |
| C191-15 | O127:H4 | O127:H4 | 131 | H27 | *aap,aar,aatA,agg3B,agg3C,agg3D,agg5A, aggR,gad,iss,ORF3,ORF4, fyuA, traT, papC, papA, hlyA, papGII, iutA, agn43* | V | B2 | 13 | Left | UNK | Yes | M |
| C193-15 | O25:H4 | O25:H4 | 131 | H27 | *aap,aar,aatA,agg3B,agg3C,agg3D,agg5A, aggR,gad,iha,iss,ORF3,ORF4,sat, fyuA, traT, papC, papA, hlyA, hlyC, aer, papGII, iutA, agn43* | V | B2 | 6 | Alive | Yes | Yes | M |
| C197-15 | O25:H- | O25:H4• | 131 | H27 | *aap,aar,aatA,agg3B,agg3C,agg3D,agg5A, aggR,cnf1,iss,ORF3,ORF4, fyuA, traT, papC, papA, hlyA, hlyC, aer, papGII, iutA, agn43* | V | B2 | 16 | Alive | No | Yes | M |
| C198-15 | O25:H4 | O25:H4 | 131 | H27 | *aap,aar,aatA,agg3B,agg3C,agg3D,agg5A, aggR,cnf1,gad,iss,ORF3,ORF4, fyuA, traT, papC, papA, hlyA, hlyC, aer, papGII, iutA, agn43* | V | B2 | 5 | Alive | No | Yes | M |
| C199-15 | O25:H4 | O25:H4 | 131 | H27 | *aap,aar,agg3B,agg3C,agg3D,agg5A,aggR, cnf1,gad,iss,ORF3,ORF4, fyuA, traT, papC, papA, hlyA, hlyC, aer, papGII, iutA, agn43* | V | B2 | 3 | Alive | No | Yes | M |
| C200-15 | O51:H4 | O51:H4 | 131 | H27 | *aap,aar,aatA,agg3B,agg3C,agg3D,agg5A, aggR,cnf1,gad,iss,ORF3,ORF4, fyuA, traT, papC, papA, hlyA, hlyC, papGII, iutA, agn43* | V | B2 | 10 | Alive | No | No | M |
| C204-15 | O86:H4 | O86:H4 | 131 | H27 | *aap,aar,aatA,agg3B,agg3C,agg3D,agg5A, aggR,cnf1,gad,iss,ORF3,ORF4, fyuA, traT, papC, papA, hlyA, hlyC, aer, papGII, iutA* | V | B2 | 7 | Dead | No | Yes | M |
| C206-15 | Orough:H4 | O25:H4 | 131 | H27 | *aap,aar,aatA,agg3B,agg3C,agg3D,agg5A, aggR,cnf1,gad,iss,ORF3,ORF4, fyuA, traT, papC, papA, hlyA, hlyC, aer, papGII, iutA, agn43* | V | B2 | 7 | Alive | Yes | Yes | M |
| C207-15 | O25:H4 | O25:H4 | 131 | H27 | *aap,aar,aatA,agg3B,agg3C,agg3D,agg5A, aggR,gad,iha,iss,mchB,mchC,mchF, ORF3,ORF4,sat* | V | B2 | 1 | Alive | No | Yes | M |
| C209-15 | O86:H4 | O86:H4 | 131 | H27 | *aap,aar,agg3B,agg3C,agg3D,agg5A,aggR, gad,iss,ORF4, fyuA, traT, papC, papA, hlyA, hlyC, aer, papGII, iutA* | V | B2 | 21 | Alive | Yes | No | M |

(*Continued*)

**Table 1.** (*Continued*)

| C-number | Phenotypical serotype | Molecular serotype | ST | Fimtype | Virulence genes | AAF | Phylogeny | Age* | Outcome | Malaria | Diarrhea | Sex |
|---|---|---|---|---|---|---|---|---|---|---|---|---|
| C210-15 | O?:H4 | O51:H4• | 131 | H27 | aap,aar,aatA,agg3B,agg3C,agg3D,agg5A, aggR,gad,iss,ORF3,ORF4, fyuA, traT, papC, papA, hlyA, hlyC, aer, papGII, iutA, agn43 | V | B2 | 15 | Alive | UNK | No | M |
| C211-15 | O25:H4 | O25:H4 | 131 | H27 | aap,aar,aatA,agg3B,agg3C,agg3D,agg5A, aggR,cnf1,gad,iss,ORF3,ORF4, fyuA, traT, papC, papA, hlyA, hlyC, aer, papGII, iutA, agn43 | V | B2 | 4 | Dead | Yes | Yes | M |
| C212-15 | O25:H4 | O25:H4 | 131 | H27 | aap,aar,aatA,agg3B,agg3C,agg3D,agg5A, aggR,gad,iss,ORF3,ORF4, fyuA, traT, papC, papA, hlyC, aer, papGII, iutA, agn43 | V | B2 | 10 | Alive | Yes | Yes | M |
| C213-15 | O25:H4 | O25:H4 | 131 | H27 | aap,aar,aatA,agg3B,agg3C,agg3D,agg5A, aggR,cnf1,gad,iss,ORF3,ORF4, fyuA, traT, papC, papA, hlyA, hlyC, aer, papGII, iutA, agn43 | V | B2 | 8 | Alive | Yes | No | M |
| C215-15 | Orough:H4 | O127:H4 | 131 | H27 | aap,aar,aatA,agg3B,agg3C,agg3D,agg5A, aggR,cnf1,gad,iss,ORF3,ORF4, fyuA, traT, papC, papA, hlyA, aer, papGII, iutA, agn43 | V | B2 | 22 | Alive | No | No | F |
| C216-15 | O127:H4 | O127:H4 | 131 | H27 | aap,aar,aatA,agg3B,agg3C,agg3D,agg5A, aggR,cnf1,gad,iss,ORF3,ORF4, fyuA, traT, papC, papA, hlyA, hlyC, aer, papGII, iutA | V | B2 | 10 | Alive | Yes | No | M |
| C217-15 | O18ac:H4 | O18ac:H4 | 131 | H27 | aap,aar,aatA,agg3B,agg3C,agg3D,agg5A, aggR,cnf1,gad,iss,ORF3,ORF4, fyuA, traT, papC, papA, hlyA, papGII, iutA, agn43 | V | B2 | 1 | Transf. | No | No | M |
| C218-15 | Orough:H4 | O25:H4 | 131 | H27 | aap,aar,aatA,agg3B,agg3C,agg3D,agg5A, aggR,cnf1,gad,iss,ORF3,ORF4, fyuA, traT, papC, papA, papGII, iutA, agn43 | V | B2 | 7 | Alive | No | Yes | M |
| C222-15 | O25:H4 | O25:H4 | 131 | H27 | aap,aar,aatA,agg3B,agg3D,agg5A,aggR, gad,iha,iss,ORF4,sat, fyuA, traT, papC, papA, hlyA, hlyC, aer, iutA, agn43 | V | B2 | 0 | Alive | No | Yes | M |
| C223-15 | O51:H4 | H4• | 131 | H27 | aap,aar,aatA,agg3B,agg3C,agg3D,agg5A, aggR,gad,iss,ORF3,ORF4, fyuA, traT, papC, papA, hlyA, hlyC, aer, papGII, iutA, agn43 | V | B2 | 14 | Alive | No | No | M |
| C165-15 | O25:H4 | O25:H4 | 131 | Neg. | aap,aar,aatA,agg3B,agg3C,agg3D,agg5A, aggR,cnf1,gad,iss,ORF3,ORF4, fyuA, traT, papC, papA, hlyA, hlyC, aer, papGII, iutA, agn43 | V | B2 | 13 | Alive | Yes | Yes | M |
| C202-15 | O25:H4 | O25:H4 | 131 | Neg. | aap,aar,aatA,agg3B,agg3C,agg3D,agg5A, aggR,cnf1,gad,iss,ORF3,ORF4, fyuA, traT, papC, papA, hlyA, aer, papGII, iutA, agn43 | V | B2 | 7 | Alive | UNK | Yes | M |
| C205-15 | O166:H15 | O166:H15 | 349 | H93 | aap,aar,aatA,agg3B,agg3C,agg3D,agg5A, aggR,air,capU,eilA,gad,iha,iss,ORF3, ORF4,sat, traT, papC, papA, hlyA, hlyC, aer, papGII, iutA, agn43 | V | D | 5 | Alive | Yes | No | M |

(*Continued*)

**Table 1.** (Continued)

| C-number | Phenotypical serotype | Molecular serotype | ST | Fimtype | Virulence genes | AAF | Phylogeny | Age* | Outcome | Malaria | Diarrhea | Sex |
|---|---|---|---|---|---|---|---|---|---|---|---|---|
| C219-15 | O166:H15 | O166:H15 | 349 | H93 | *aap,aar,aatA,agg3B,agg3C,agg3D,agg5A, aggR,air,capU,eilA,gad,iha,iss,*ORF3, ORF4,*sat, traT, papC, papA, hlyA, hlyC, aer, papGII, iutA, agn43* | V | D | 8 | Dead | Yes | No | F |

*Age in months; UNK–unknown; and REF- referred to Maputo central hospital; Transf.—Transferred

Definitions: *papA*, P fimbriae structural subunit; *papC*, P fimbria assembly; *papGII*, P fimbria adhesin; *fimH*, type 1 fimbriae; *hra*, heat-resistant agglutinin; *hlyA*, α-hemolysin; *hlyC* hemolysin, *cnf1*, cytotoxic necrotizing factor; *sat*, secreted autotransporter toxin; *pic*, autotransporter protease; *fyuA*, yersiniabactin system; *aer*, aerobactin; *iutA*, aerobactin receptor; *traT*, serum resistance associated; *ibeA*, invasion of brain endothelium; *astA*, EAEC heat-stable toxin; *pet*, Plasmid-encoded toxin; *sigA*, IgA protease-like homolog; *pic* Serine protease precursor; *sepA* Shigella extracellular protease; ORF3 ORF4—Cryptic protein; *aap*, Dispersin, antiaggregation protein; *aaiC*, AaiC secreted protein; *aggR*, EAEC transcriptional activator; *aatA*, Dispersin transporter protein; *aggA*, AAF/I major fimbrial subunit; *aggB*, minor fimbrial subunit; *aggC*, usher; *aggD*, chaperone; *agg4A*, AAF/IV major fimbrial subunit; *agg4B*, minor fimbrial subunit; *agg4C*, usher; *agg4D*, chaperone; *agg5A*, AAF/V fimbrial subunit; *agg3B*, minor fimbrial subunit; *agg3C*, usher; *agg3D*, chaperone; *aar*, AggR-activated regulator; *eilA*, *Salmonella* HilA homolog; *capU*, Hexosyltransferase homolog; *air*, Enteroaggregative immunoglobulin; *iss*, increased serum survival gene, *agn43*; outer membrane protein; *iha*, adherence-conferring protein; *gad*, Glutamate decarboxylase; *mchB*, *mchC*, *mcmA*, genes required for the production of the antimicrobial peptide microcin H47; *mchF*; microcin transporter protein; *nfaE*, non-fimbrial adhesion the molecular serotype confirmed the conventional serotype except in seven strains (marked with •)

comparing EAEC and other bacteremic *E. coli* isolates analyzed by conventional PCR. Notably, 35 out of 44 (79.5%) of EAEC isolates contained the three genes—*iutA*, *fyuA* and *traT* (associated with ExPEC) similar to 65% (100 isolates) of other bacteremic *E. coli*. The toxins commonly found in septicemic *E. coli* (e.g. *hlyA*, *hylC* and *cnf1*) were more prevalent among EAEC isolates compared to other bacteremic *E. coli* (Tables 2 and 3).

**Table 2. Comparison of virulence factors of EAEC causing bacteraemia *versus* other bacteremic *E. coli* strains isolated in Mozambican children detected by conventional multiplex PCR–EAEC associated virulence genes.**

| Virulence Gene | Bacteraemic EAEC (N = 44) n (%) | Other bacteraemic *E. coli* (N = 164) n (%) | p-value |
|---|---|---|---|
| EAEC | | | |
| *aatA* | 39 (88.6) | 0 (0.0) | <0.001 |
| *aaiC* | 2 (4.6) | 0 (0.0) | 0.044 |
| *aggR* | 41 (93.2) | 0 (0.0) | <0.001 |
| *aap* | 41 (**93.2**) | 12 (7.3) | <0.001 |
| ORF3 | 39 (88.6) | 10 (6.1) | <0.001 |
| ORF4 | 41 (93.2) | 6 (3.7) | <0.001 |
| *aar* | 41 (93.2) | 44 (27.3) | <0.001 |
| *astA* | 5 (11.4) | 72 (43.9) | <0.001 |
| EAEC adhesins | | | |
| *aafC* | 0 (0.0) | 22 (13.4) | 0.005 |
| *agg3/4/5C* | 38 (88.4) | 7 (4.2) | <0.001 |
| *aggA* | 2 (4.5) | 2 (1.2) | NS |
| *agg3A* | 0 (0.0) | 0 (0.0) | NS |
| *aafA* | 0 (0.0) | 3 (1.8) | NS |
| *agg4A* | 1 (2.3) | 21 (12.8) | 0.052 |
| Agg5A | 39 (88.6) | 7 (4.3) | <0.001 |

**Table 3. Comparison of virulence factors of EAEC causing bacteraemia *versus* other bacteremic *E. coli* isolated in Mozambican children detected by conventional multiplex PCR–ExPEC and miscellaneous associated virulence genes.**

| Virulence Gene | Bacteremic EAEC (N = 44) n (%) | Other bacteremic *E. coli* (N = 164) n (%) | p-value |
|---|---|---|---|
| ExPEC adhesins | | | |
| *sfaS* | 0 (0.0) | 31 (18.9) | 0.002 |
| *fimH* | **41** (93.**2**) | **153** (93.3) | NS |
| *papA* | 43 (97.7) | 96 (58.5) | <0.001 |
| *papC* | 37 (84.1) | 132 (80.5) | NS |
| *papGII* | 38 (86.4) | 125 (76.2) | 0.033 |
| *papGIII* | 1 (2.3) | 3 (1.8) | NS |
| *afa_dra* | 3 (6.8) | 63 (38.4) | <0.001 |
| Class I SPATEs | | | |
| *sat* | 1 (22.7) | 31 (18.9) | NS |
| *pet* | 0 (0.0) | 32 (19.5) | 0.001 |
| *sigA* | 0 (0.0) | 1 (0.6) | NS |
| Class II SPATEs | | | |
| *sepA* | 1 (2.3) | 19 (11.6) | NS |
| *pic* | 4 (9.1) | 15 (9.2) | NS |
| *vat* | 3 (6.8) | 59 (36.0) | <0.001 |
| *epeA* | 0 (0.0) | 0 (0.0) | NS |
| *eatA* | 0 (0.0) | 3 (1.8) | NS |
| Siderophores | | | |
| *iutA* | 41 (93.2) | 144 (87.8) | NS |
| *iroN* | **0** (0.0) | **67** (41.1) | <0.001 |
| *fyuA* | 40 (90.1) | 137 (83.5) | 0.043 |
| *aer* | 30 (68.2) | 127 (77.4) | NS |
| Toxins | | | |
| *hlyA* | 36 (81.8) | 50 (30.4) | <0.001 |
| *hlyC* | 31 (70.5) | 70 (42.7) | <0.001 |
| *cnf1* | 27 (61.4) | 58 (35.4) | 0.002 |
| *hra* | 4 (9.1) | 47 (28.6) | 0.007 |
| *cdtb* | 0 (0.0) | 0 (0.0) | NS |
| *agn43* | 38 (86.4) | 26 (15.9) | 0.017 |
| Miscellaneous | | | |
| *air* | 4 (9.1) | 33 (20.1) | NS |
| *eilA* | 5 (11.4) | 85 (52.4) | <0.0001 |
| *capU* | 5 (11.4) | 35 (21.6) | NS |
| *ibeA* | 0 (0.0) | 11 (6.7) | NS |
| *traT* | 38 (86.4) | 131 (79.9) | NS |

## Whole genome sequencing (WGS) of EAEC isolates

WGS data found that the 44 EAEC strains belonged to six different ST types with ST-131 accounting for 84.1% (37/44) of the strains, of which 35 (95.6%) harbor *fimH27*, while the other two were negative for *fimH* typing. Furthermore, 59% (22/37) ST131 strains belonged to the serotype O25:H4, while the remaining 41% (15/37) fall into five serotypes [O127:H4 (5 strains), O86:H4 (4), O51:H4 (4), O18ac:H4 (1) and O15:H4 (1)] (Table 1). Additionally, 93.2% (41/44) harbored the EAEC virulence plasmid pAA encoding the master regulator *aggR* and several *aggR*- regulated genes, such as *aap* (dispersin), *aatA* (dispersin translocator), the

newly discovered *aggR* repressor *aar* [32], and the aggregative adherence fimbriae (AAF) gene cluster. The AAF/V variant was found in 88.6% (39/44) of the EAEC strains followed by AAF/I (2/44) and AAF/IV (1/44). The sequence analysis confirmed the PCR analysis and the presences of several ExPEC associated virulence genes (*iutA, fyuA, traT, hlyA, hylC, cnf1, and pap-GII*). Furthermore, we found the increased serum survival gene—*iss* in 95.5% (42/44) of the EAEC strains positive for *aggR*. Lastly, the molecular serotype confirmed the conventional serotype except in seven strains (marked with • in Table 1). Additionally, virotype classification did not provide a clear discrimination of our strains according to Dahbi et. al. [33], suggesting possible occurrence of virotype E sub-type requiring further characterization of isolates.

## Antimicrobial susceptibility and associated mechanisms

We also assessed the antimicrobial resistance of the 44 EAEC strains, documenting high prevalence of resistance to the most commonly available and used antibiotic for empirical treatment (ampicillin, gentamicin and chloramphenicol) in our community including multidrug resistance (MDR) 97% as demonstrated in Table 4. WGS also identified genes conferring resistance towards three or more groups of antibiotics; Aminoglycosides, Macrolides, Phenicols, Quinolones, Sulphonamides, Tetracyclines, Trimethoprim, and/or β-Lactams (Table 5).

## Burden and clinical impact of EAEC bacteremia

Of the total of 325 *E. coli* bacteremia episodes, only 127 (39.1%) occurred in children living within the DSS area, yielding an overall incidence rate of *E. coli* of 110.1 cases/100,000 child-years (95%CI: 92.5–131.0) with the highest incidence occurring among infants (181.1 cases/100,000 child-years; 95%CI: 143.7–228.1) (Table 6). EAEC incidence among infants was 45.3

**Table 4. Antimicrobial susceptibility profile of EAEC causing bacteremia in young children in Manhiça District Hospital.**

| Antibiotic name | No. tested | %R | %I | %S |
|---|---|---|---|---|
| Ampicillin | 33 | 97 | 0 | 3 |
| Amoxicillin/Clavulanic acid | 33 | 21,2 | 33,3 | 45,5 |
| Piperacillin/Tazobactam | 31 | 0 | 0 | 100 |
| Cefuroxime | 33 | 0 | 36,4 | 63,6 |
| Ceftazidime | 33 | 0 | 0 | 100 |
| Ceftriaxone | 33 | 0 | 0 | 100 |
| Cefoxitin | 33 | 0 | 3 | 97 |
| Aztreonam | 33 | 3 | 0 | 97 |
| Ertapenem | 33 | 0 | 3 | 97 |
| Imipenem | 33 | 3 | 0 | 97 |
| Meropenem | 33 | 0 | 0 | 100 |
| Amikacin | 33 | 6,1 | 0 | 93,9 |
| Gentamicin | 33 | 48,5 | 0 | 51,5 |
| Tobramycin | 33 | 24,2 | 6,1 | 69,7 |
| Nalidixic acid | 33 | 3 | 3 | 93,9 |
| Ciprofloxacin | 33 | 18,2 | 0 | 81,8 |
| Trimethoprim/Sulfamethoxazole | 33 | 81,8 | 0 | 18,2 |
| Chloramphenicol | 33 | 72,7 | 0 | 27,3 |
| Tetracycline | 33 | 39,4 | 0 | 60,6 |

**Table 5. Detection by WGS of genes conferring resistance of EAEC causing bacteremia in young children in Manhiça District Hospital.**

| C-number | ST | Molecular serotype | fimtype | Phylogeny | Resistance |
|---|---|---|---|---|---|
| C159-15 | 131 | O25:H4 | *fimH27* | B2 | aac(3)-IId-like,blaTEM-1B,catA1-like,dfrA7,strA,strB,sul1-like,sul2 |
| C164-15 | 131 | O25:H4 | *fimH27* | B2 | aac(3)-IId-like,blaTEM-1B,catA1-like,dfrA7,strA,strB,sul1-like,sul2,tet(A) |
| C165-15 | 131 | O25:H4 | Negative | A | aac(3)-IId-like,blaTEM-1B,catA1-like,dfrA7,strA,strB,sul1-like,sul2 |
| C166-15 | 131 | O127:H4 | *fimH27* | B1 | aac(3)-IId-like,blaTEM-1B,catA1-like,dfrA7,strA,strB,sul1-like,sul2,tet(A) |
| C167-15 | 131 | O25:H4 | *fimH27* | B1 | aac(3)-IId-like,blaTEM-1B,catA1-like,dfrA7,strA,strB,sul1-like,sul2 |
| C168-15 | 131 | O51:H4 | *fimH27* | B1 | aac(3)-IId-like,blaTEM-1B,catA1-like,dfrA7,strA,strB,sul1-like,sul2,tet(A) |
| C169-15 | 10 | O176:H33 | *fimH54* | A | aadA1-like,blaOXA-1-like,catA1-like,strA,strB,sul2,tet(B) |
| C170-15 | 131 | O25:H4 | *fimH27* | B2 | aac(3)-IId-like,blaTEM-1B,catA1-like,dfrA7,strA,strB,sul1-like,sul2 |
| C171-15 | 131 | O25:H4 | *fimH27* | B2 | aac(3)-IId-like,aadA2,blaTEM-1B,catA1-like,dfrA12,mph(A),strA,strB-like,sul1,sul2 |
| C174-15 | 131 | O86:H4 | *fimH27* | B1 | aac(3)-IId-like,blaTEM-1B,catA1-like,dfrA7,strA,strB,sul1-like,sul2 |
| C175-15 | 130 | O17/O77:H34 | *fimH47* | D | Not found |
| C177-15 | 131 | O25:H4 | *fimH27* | B1 | aac(3)-IId-like,blaTEM-1B,catA1-like,dfrA7,strA,strB,sul1-like,sul2 |
| C178-15 | 131 | O25:H4 | *fimH27* | B1 | aac(3)-IId-like,blaTEM-1B,catA1-like,dfrA7,strA,strB,sul1-like,sul2 |
| C179-15 | 131 | O127:H4 | *fimH27* | A | aac(3)-IId-like,blaTEM-1B,catA1-like,dfrA7,strA,strB,sul1-like,sul2,tet(A) |
| C182-15 | 131 | O15:H4 | *fimH27* | B2 | aac(3)-IId-like,blaTEM-1B,catA1-like,dfrA7,strA,strB,sul1-like,sul2 |
| C184-15 | 131 | O25:H4 | *fimH27* | B2 | aac(3)-IId-like,blaTEM-1B,catA1-like,dfrA7,strA,strB,sul1-like,sul2 |
| C185-15 | 131 | O25:H4 | *fimH27* | B2 | aac(3)-IId-like,blaTEM-1B,catA1-like,dfrA7,strA,strB,sul1-like,sul2 |
| C188-15 | 38 | H30 | *fimH5* | D | aadA1,blaTEM-1B,catA1-like,dfrA1,strA,strB,sul2,tet(D) |
| C189-15 | 131 | O86:H4 | *fimH27* | B2 | aac(3)-IId-like,blaTEM-1B,catA1-like,dfrA7,strA,strB,sul1-like,sul2 |
| C190-15 | 31 | O11:H18 | Negative | A | blaTEM-1B,catA1-like,dfrA7,strA,strB,sul1-like,sul2,tet(A) |
| C191-15 | 131 | O127:H4 | *fimH27* | B1 | aac(3)-IId-like,blaTEM-1B,catA1-like,dfrA7,strA,strB,sul1-like,sul2 |
| C193-15 | 131 | O25:H4 | *fimH27* | B1 | aac(3)-IId-like,aadA2,blaTEM-1B,catA1-like,dfrA12,dfrA14-like,mph(A),strA-like,strB,sul1,sul2 |
| C194-15 | 10 | O176:H33 | *fimH54* | A | aadA1-like,blaOXA-1-like,catA1-like,strA,strB,sul2,tet(B) |
| C197-15 | 131 | O25:H4 | *fimH27* | B1 | aac(3)-IId-like,blaTEM-1B,catA1-like,dfrA7,strA,strB,sul1-like,sul2 |
| C198-15 | 131 | O25:H4 | *fimH27* | B1 | aac(3)-IId-like,blaTEM-1B,catA1-like,dfrA7,strA,strB,sul1-like,sul2 |
| C199-15 | 131 | O25:H4 | *fimH27* | B1 | aac(3)-IId-like,blaTEM-1B,catA1-like,dfrA7,strA,strB,sul1-like,sul2 |
| C200-15 | 131 | O51:H4 | *fimH27* | B1 | aac(3)-IId-like,blaTEM-1B,catA1-like,dfrA7,strA,strB,sul1-like,sul2 |
| C202-15 | 131 | O25:H4 | Negative | B2 | aac(3)-IId-like,blaTEM-1B,catA1-like,dfrA7,strA,strB,sul1-like,sul2 |
| C204-15 | 131 | O86:H4 | *fimH27* | B2 | aac(3)-IId-like,blaTEM-1B,catA1-like,dfrA7,strA,strB,sul1-like,sul2 |
| C205-15 | 349 | O166:H15 | *fimH93* | B1 | blaTEM-1B,strA,strB-like,sul2 |
| C206-15 | 131 | O25:H4 | *fimH27* | D | aac(3)-IId-like,blaTEM-1B,catA1-like,dfrA7,strA,strB,sul1-like,sul2 |
| C207-15 | 131 | O25:H4 | *fimH27* | B2 | aac(3)-IId-like,aadA2,blaTEM-1B,catA1-like,dfrA12,mph(A),sul1 |
| C209-15 | 131 | O86:H4 | *fimH27* | B1 | aac(3)-IId-like,blaTEM-1B |
| C210-15 | 131 | O51:H4 | *fimH27* | B1 | aac(3)-IId-like,blaTEM-1B,catA1-like,dfrA7,strA,strB,sul1-like,sul2,tet(A) |
| C211-15 | 131 | O25:H4 | *fimH27* | B1 | aac(3)-IId-like,blaTEM-1B,catA1-like,dfrA7,strA,strB,sul1-like,sul2 |
| C212-15 | 131 | O25:H4 | *fimH27* | B1 | aac(3)-IId-like,blaTEM-1B,catA1-like,dfrA7,strA,strB,sul1-like,sul2 |
| C213-15 | 131 | O25:H4 | *fimH27* | B1 | aac(3)-IId-like,blaTEM-1B,catA1-like,dfrA7,strA,strB,sul1-like,sul2 |
| C215-15 | 131 | O127:H4 | *fimH27* | D | aac(3)-IId-like,blaTEM-1B,catA1-like,dfrA7,strA,strB,sul1-like,sul2 |
| C216-15 | 131 | O127:H4 | *fimH27* | A | aac(3)-IId-like,blaTEM-1B,catA1-like,dfrA7,strA,strB,sul1-like,sul2 |
| C217-15 | 131 | O18ac:H4 | *fimH27* | B1 | aac(3)-IIa,aac(3)-IId-like,blaTEM-131-like,blaTEM-1B,catA1-like,dfrA7,floR-like,strA,strB,sul1-like,sul2,tet(A)-like |
| C218-15 | 131 | O25:H4 | *fimH27* | B1 | aac(3)-IId-like,blaTEM-1B,catA1-like,dfrA7,strA,strB,sul1-like,sul2 |
| C219-15 | 349 | O166:H15 | *fimH93* | A | blaTEM-1B,dfrA8,strA,strB-like,sul2,tet(A) |
| C222-15 | 131 | O25:H4 | *fimH27* | B2 | aac(3)-IId-like,aadA2,blaTEM-1B,catA1-like,dfrA12,sul1 |
| C223-15 | 131 | :H4 | *fimH27* | B2 | aac(3)-IId-like,blaTEM-1B,catA1-like,dfrA7,strA,strB,sul1-like,sul2,tet(A) |

**Table 6. Minimum community-based incidence rates of *Escherichia coli* bacteemia among children aged less than 3 years living in Manhiça DSS study area, 2001–2014.**

| Category | Time-at-risk | No. episodes | Incidence rates [a] (95% CI) |
|---|---|---|---|
| Bacteremic EAEC | | | |
| 0–11 months | 39767.38 | 18 | 45.3 (28.5–71.8) |
| 12–23 months | 38283.89 | 7 | 18.3 (8.7–38.4) |
| 24–35 months | 37309.93 | 3 | 8.04 (2.6–24.9) |
| All ages | 115361.20 | 28 | 24.3 (16.7–35.2) |
| Other *E. coli* bacteremia | | | |
| 0–11 months | 39766.1 | 54 | 135.8 (104.0–177.3) |
| 12–23 months | 38282.7 | 37 | 96.7 (70.0–133.4) |
| 24–35 months | 37309.7 | 8 | 21.4 (10.7–42.9) |
| All ages | 115358.8 | 99 | 85.8 (70.5–104.5) |
| All bacteremic *E. coli* | | | |
| 0–11 months | 39765.5 | 72 | 181.1 (143.7–228.1) |
| 12–23 months | 38282.5 | 44 | 114.94 (85.5–154.5) |
| 24–35 months | 37309.6 | 11 | 29.5 (16.3–53.2) |
| All ages | 115357.5 | 127 | 110.1 (92.5–131.0) |

[a] Incidence rates were calculated referring cases to population denominators establishing time at risk (child years at risk [CYAR]) inferred from the HDSS census information. Children did not contribute to the numerator or denominator for a period of 15 days after each episode or when they were outside the study area.

cases/100,000 child-years (95%CI: 28.5–71.8), peaking in 2002 and 2003 with 71.5 and 78.9 cases/100,000 children-years, respectively (Fig 1).

Nutritional status was recorded for 51 of the 57 EAEC patients, of whom 8 (15.7%) were severely malnourished, compared with 70/228 (30.7%) from other *E. coli* bacteremia group, p = 0.031; and the CFR was similar in the two groups (15.8%, 9/57 *vs.* 14.7%, 38/258; p = 0.8, for EAEC and other *E. coli* bacteremia, respectively). Fever (92.9% *vs.* 89.5%), diarrhea (42.1% *vs.* 32.5%), vomiting (36.8% *vs.* 29.1%), and cough (71.9% *vs.* 73.3%) were found in similar proportion between children infected with EAEC (n = 57) and other *E. coli* bacteremia (n = 258), respectively.

The Classification and Regression Tree (CART) analysis suggests the presence of 2 clusters associated with poor outcome in the absence of malaria. Cluster 1 comprising strains testing positive for *papGII* and *hra* in the absence of *sfaS* (Node 1) and cluster 2 comprising strains harboring *cnf1* in the absence of *hra* and *afa_dr* (Node 2) (Fig 2). In addition, we demonstrated the presence of fatal strains harbored *hlyC* and *orf3* genes in the absence of *agn43* (Node 1) or belonging to ST131 clone harboring *hlyA* and *aer* lacking *astA* toxin (Node 2) among children infected by EAEC (Fig 3). Case fatality ratio among children infected sequenced EAEC strains was 14.6% (6/41), mostly related to ST131 strains (83.3%; 5/6); and 60% (3/5) children with poor outcome were infected by serotypes other than O25:H4, namely O86:H4 (n = 2) and O127:H4 (n = 1).

Classification and regression tree (CART) classification tree topology reveals combinations of factors most strongly associated death in the absence of malaria (Fig 2) or for EAEC strains (Fig 3). We considered all genotypic and phenotypic assays performed: *aatA, aggR, aaiC, aap, ORF3, sat, sepA, pic, sigA, pet, astA, aafC, agg3/4C, aafA, agg3A, aggA, agg4A, air, capU, eilA, ORF61*. Each branch of the CART tree ends in a terminal "node" (blue boxes), and each terminal node is uniquely defined by the presence or absence of a predictive factor such as a gene. The tree is hierarchical in nature.

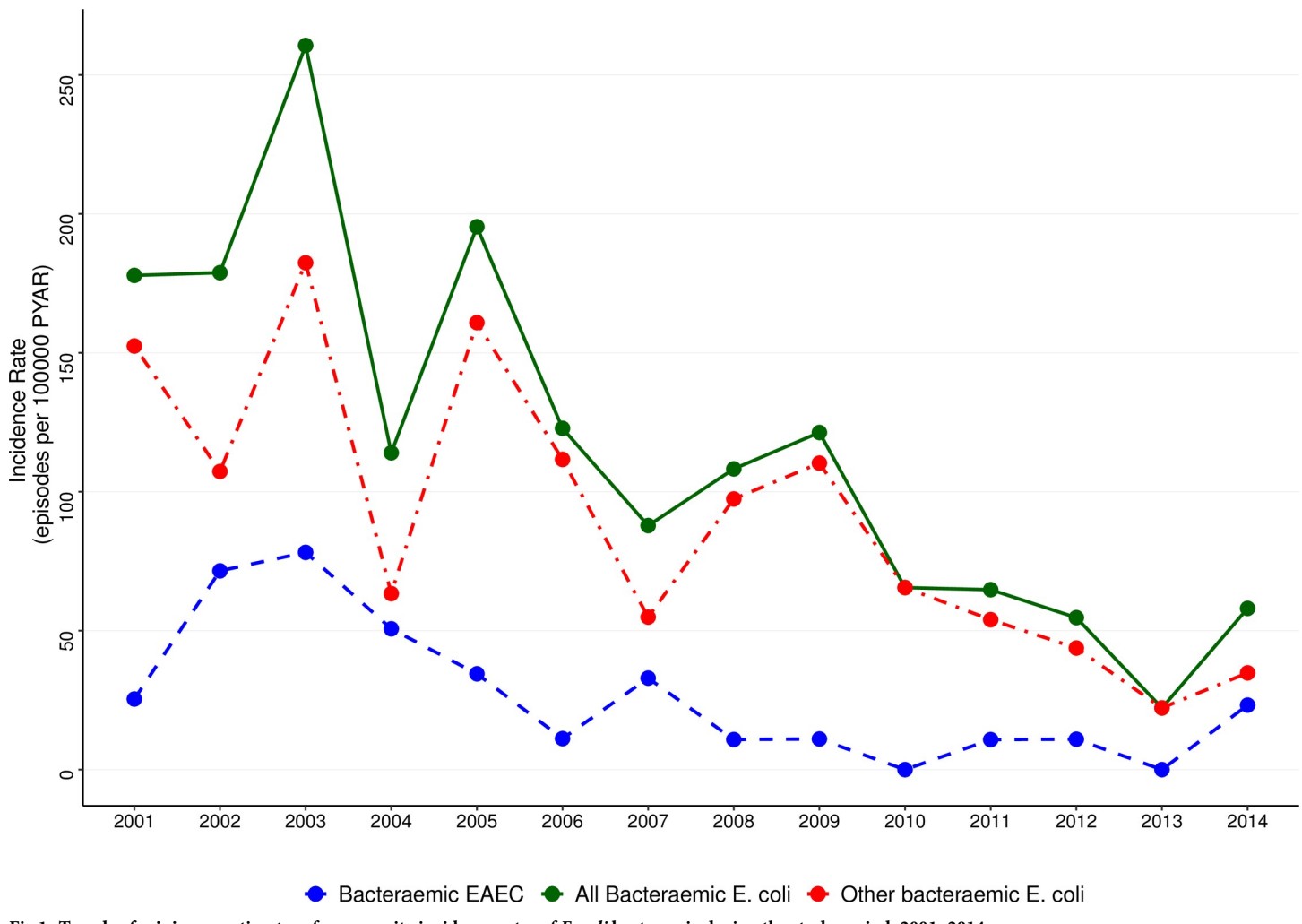

**Fig 1. Trends of minimum estimates of community incidence rates of *E. coli* bacteremia during the study period, 2001–2014.**

## Discussion

This is the first study conducted in Mozambique characterizing *E. coli* strains causing childhood bacteremia; documenting the novel subclone of ST131 harboring EAEC genes causing bacteremia in children, with its highest incidence peaking during infancy. During the last years, evidence of involvement of non-*fimH30* ST131 isolates and *fimH30* subclone isolates fulfilling molecular criteria for EAEC in extra-intestinal infections have been reported [18,34]. However, to our knowledge, this is the first report analyzing overtime trend incidences of EAEC-associated bacteremia in African children, showing a magnitude similar to that caused by *Staphylococcus aureus* or Hib (pre-vaccine introduction) in our population [6], suggesting that EAEC is currently playing an important role as a cause of childhood *E. coli* bacteremia in this setting.

The high incidence of EAEC reported here could either be related to pathogen or to host factors, including malnutrition or HIV, both highly prevalent in our study area and also known to enhance translocation of commensal bacteria to the bloodstream [2]. Despite the limitation on HIV data in our study population, we believe that the EAEC incidence reported here is possibly due to properties of the pathogen. If it was favored by HIV or malnutrition co-

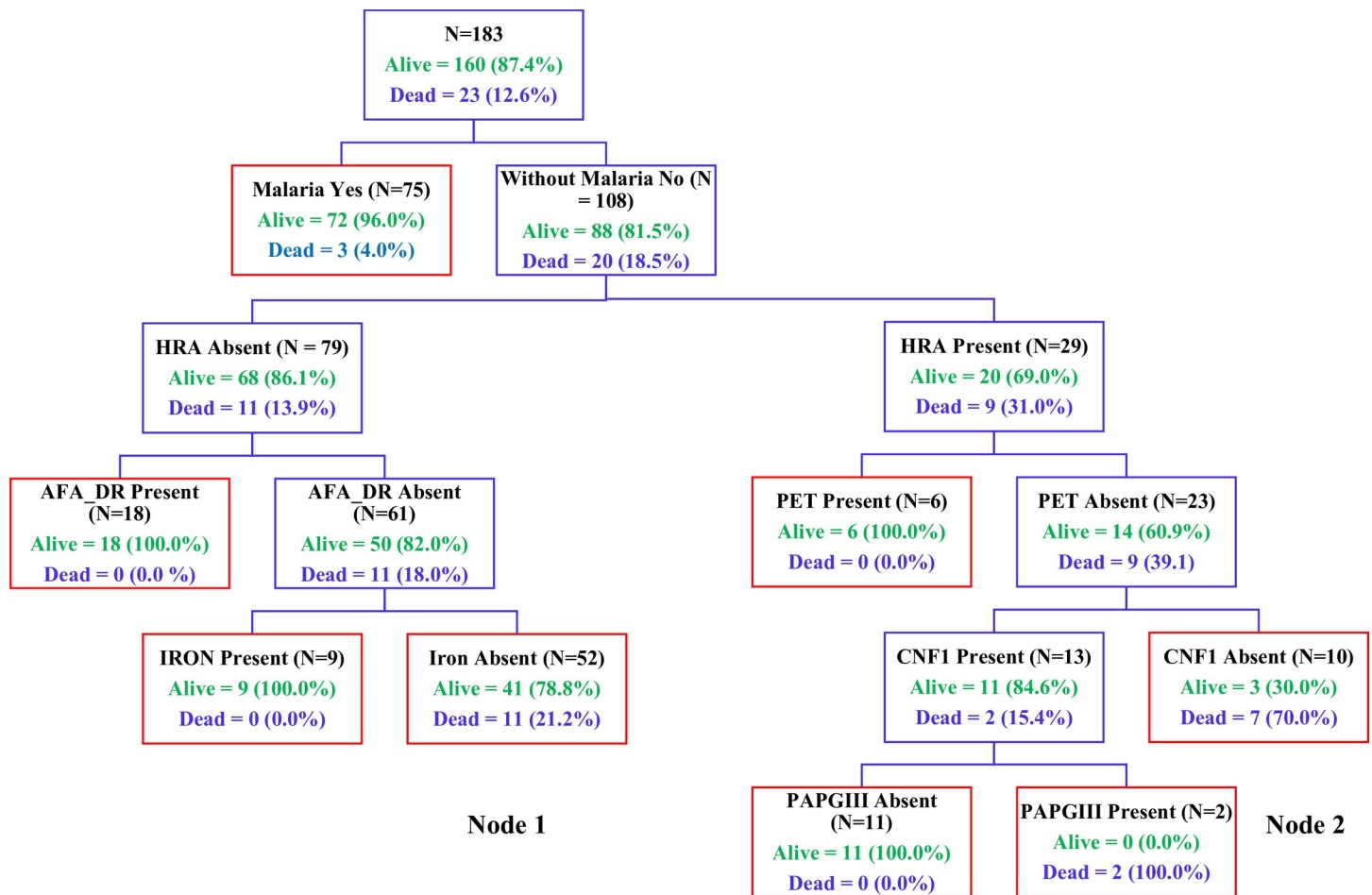

**Fig 2. Diagram of CART analysis against hospital outcome.** We included the collective number of virulence genes present (virulence factor score, VFS) in putting 48 factors of interest as binary (present/absent) independent predictive variables along with a continuous "factor total" that was a sum of all factors including the presence of malaria. We identified 2 clusters associated with poor outcome in the absence of malaria: i) comprising strains testing positive for *papGII* and *hra_* in the absence of *sfaS* (node 1); and ii) comprising strains harboring *cnf1* in the absence of *hra* and *afa_dr* (node 2).

infection, we would expect to also find a high prevalence of the other pathotypes (e.g. EPEC or ETEC) also prevalent among healthy children [35].

As a common enteric isolate, we hypothesize that extra-intestinal EAEC may arise via the transfer the pAA plasmid more classical invasive pathogens, thus transferring additional virulence traits. This is supported by the high prevalence of classical ExPEC virulence genes within our EAEC, such as *hlyA* which is known to induce oxidative stress in blood [25], and which is also associated with polymorphonuclear lysis/necrosis and lung injury *in vivo* in a rat model of *E. coli* pneumonia [36]. The low prevalence of the chromosomal *aaiC* gene in our strains compared to *aggR* and *aatA* on the pAA plasmid may be additional support for transfer of the pAA plasmid. It also suggests that *aaiC* is not a good marker for EAEC bacteremic strains in our community.

The high prevalence of adhesion AAF/V in our strains is noteworthy, suggesting a high degree of phylogenetic relatedness of our strains. This is underscored by the presence of *papC* or type 1 fimbriae *(fimH)* in more than 90% of isolates, suggesting that these strains may derive from urinary tract infections (UTI) strains, despite the lack of clinical information with regard to diagnosis of UTI. The change in *fimH* alleles might improve colonization abilities of the

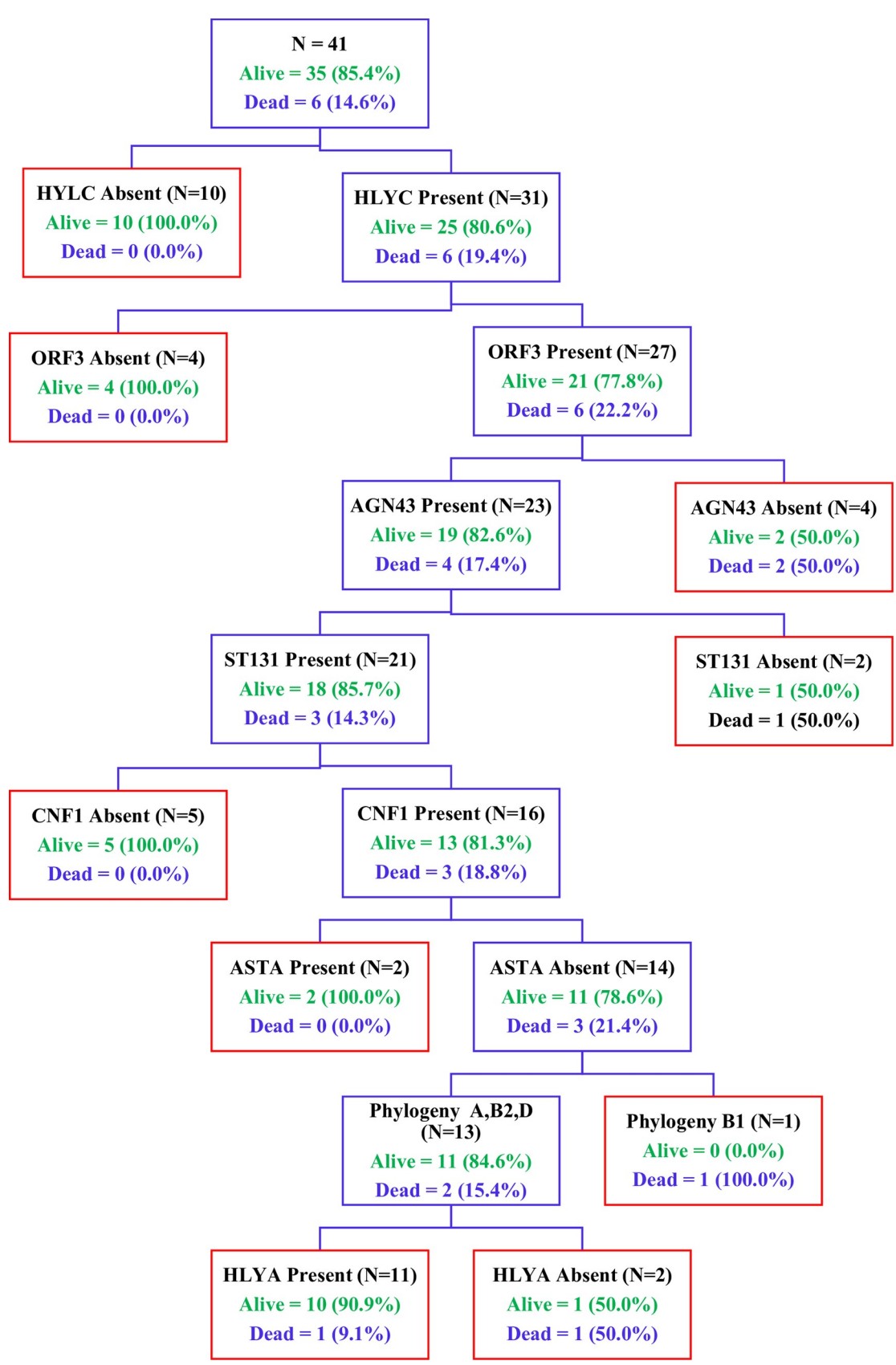

**Fig 3. CART analyses for bacteremic EAEC assessing 66 genes by WGS.** Regardless of age or serotype we demonstrated the presence of fatal strains harboring *hlyC* and *orf61* genes in the absence of *agn43* (node 1) or belonging to ST131 clone harboring, *hlyA* and *aer* lacking *astA* toxin (node 2).

different clades (global dissemination of a multidrug resistant *E. coli* clone) [33], however, to our knowledge this is second report of *fimH27* subclade, which has recently been reported in ST405 accounting for 13% of ExPEC strains isolated from clinical isolates in Nigeria [12]. In contrast to the *fimH30* subclade that is characterized to be resistant to extended spectrum of β-lactamase or fluoroquinolones [15,37,38], our strains were susceptible to third generation cephalosporines and fluoroquinolones. However, the resistance profile of our *fimH27* strains was similar to those reported by Roer et al from bloodstream infections in Denmark [39], despite the high serotype- and ST diversity in the latter. Interestingly, the fact that the isolates circulating in our community do not fit in the classical classification of virotypes [33] supports the hypothesis of the presence of a new entity that require further characterization. Plasmid analysis of our strains compared to globally disseminated *fimH30* and *fimH27* of ST405 is underway and will be published elsewhere.

Also notable is the fact that WGS identified the presence of the *iss* gene, recognized for its role in ExPEC virulence and considered a distinguishing trait of avian ExPEC but not of human ExPEC [40], suggesting that some strains may fit in the classification of avian pathogenic *E. coli* (APEC).

In addition, both serotyping and WGS data support the presence of serotype O25:H4 clone ST131, a clone significantly associated with urinary tract infections and bacteremia [37]. More interesting is the finding that 15 out of 37 (41.7%) of ST131 strains were from distinct serotypes from the traditional O25:H4 and to our knowledge never previously reported: O127:H4 (5 strains), O51:H4 (4), O86:H4 (4), O18ac:H4 and O15:H4 [41,42]. EPEC has been shown to cluster in related groups sharing the H antigen (H2 and H6) that differ only on the O antigen, which might suggest that the LPS operon may be located in a phage region and can be transferred by transduction among EPEC [43]. Here, we find that these strains share H4 but differ in their O antigen, yet belong to the same MLST type ST131, a finding that certainly warrants further investigation. Importantly, *E. coli* phylogenetic analyses have generally attached greater significance to the H antigen as a marker of shared genetic ancestry, suggesting that the high preponderance of H4 strains in Mozambique may indeed signal the existence of a highly virulent and longstanding pathogen. Further epidemiologic studies should address the importance of H4 flagellar clones.

The WGS analysis illustrated that the recently discovered Aar, that has been hypothesized to act directly or indirectly as a virulence suppressor, modulating virulence because of selection towards clinical attenuation, was present in almost all isolates. These data may also strongly support an important role for the *aar* gene in *E. coli* epidemiology; similar to what was found in previous studies in Mali and Brazil where aar-negatives AAF/IV variant showed increased pathogenicity [24,44].

Interestingly CART analysis data showed that deaths are likely to occur in the absence of malaria; nearly 20% of such children died, of which 12 when infected with strains harboring *papGII* and eight in those testing positive for *hra* (Fig 2, node 1). This finding reinforces the need of routine screening of bacterial pathogens among children admitted in developing countries where most of deaths are attributable to malaria due to the limited microbiology infrastructure. Indeed, fatal EAEC strains are also related to the presence or absence of specific virulence factors found in ST131 strains (Fig 3) with attributable case fatality greater than that caused by invasive non-typhoidal *Salmonella* [45] or *S. aureus*. In addition, despite the small

number of isolates, the poor outcome of ST131 non-O25:H4 serotypes may suggest that those are more virulent that the classical serotype O25:H4, and require further *in vitro* or *in vivo* testing to establish its potential virulence [13]. Unfortunately due to the lack of adequacy or incomplete data on appropriate empirical treatment in terms of number of doses and days, the variables of antimicrobial resistance, HIV treatment were not included in the CART analysis which may help to elucidate the relationship of strains virulence profile and poor outcome

Our study sheds light on the etiology of the bacteremia events, suggesting that not only O25:H4 EAEC, but also other previously undescribed EAEC serotypes of ST131 clone strains can cause clinically severe invasive bacteremia in neonates and young children resulting in hospitalization and death in Southern Mozambique, requiring prompt recognition for appropriate management.

## Supporting information

**S1 Table. Distribution of distribution of contig size lengths, genome size and N50s of the EAEC sequenced strains.**
(XLSX)

## Acknowledgments

The authors thank the CISM and MDH staff for collecting and processing data; and the District Health Authorities for their collaboration in the research activities on-going in the Manhiça district. Special thanks for the CISM Bacteriology and Molecular Biology laboratory technicians for sample processing. Special thanks for Augusto Messa Junior for revision of the manuscript. We are indebted to the children and mothers participating in the study.

## Author Contributions

**Conceptualization:** Inácio Mandomando, Nadia Boisen, Joaquim Ruiz, Fernando Ruiz-Perez, James P. Nataro.

**Data curation:** Inácio Mandomando, Delfino Vubil, Nadia Boisen, Joaquim Ruiz, Betuel Sigaúque, Tacilta Nhampossa, Marcelino Garrine, Sergio Massora, Pedro Aide, Ariel Nhacolo, Maria J. Pons, Quique Bassat.

**Formal analysis:** Inácio Mandomando, Nadia Boisen, Llorenç Quintó, Joaquim Ruiz.

**Funding acquisition:** Inácio Mandomando, Nadia Boisen, Pedro Aide.

**Investigation:** Inácio Mandomando, Delfino Vubil, Nadia Boisen, Llorenç Quintó, Joaquim Ruiz, Betuel Sigaúque, Tacilta Nhampossa, Marcelino Garrine, Sergio Massora, Pedro Aide, Ariel Nhacolo, Maria J. Pons, Quique Bassat, Jordi Vila, Eusébio Macete, Flemming Scheutz, Myron M. Levine, Fernando Ruiz-Perez, James P. Nataro, Pedro L. Alonso.

**Methodology:** Inácio Mandomando, Delfino Vubil, Nadia Boisen, Llorenç Quintó, Joaquim Ruiz, Marcelino Garrine, Jordi Vila, Flemming Scheutz, Fernando Ruiz-Perez, James P. Nataro.

**Project administration:** Inácio Mandomando, Pedro Aide.

**Resources:** Inácio Mandomando, Nadia Boisen, Pedro Aide.

**Supervision:** Pedro Aide, James P. Nataro.

**Validation:** Inácio Mandomando, Nadia Boisen.

**Visualization:** Inácio Mandomando, Nadia Boisen.

**Writing – original draft:** Inácio Mandomando.

**Writing – review & editing:** Inácio Mandomando, Delfino Vubil, Nadia Boisen, Llorenç Quintó, Joaquim Ruiz, Betuel Sigaúque, Tacilta Nhampossa, Marcelino Garrine, Sergio Massora, Pedro Aide, Ariel Nhacolo, Maria J. Pons, Quique Bassat, Jordi Vila, Eusébio Macete, Flemming Scheutz, Myron M. Levine, Fernando Ruiz-Perez, James P. Nataro, Pedro L. Alonso.

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
