## [Decision Letter · Decision Letter 0]

11 Feb 2020

Dear Dr. Mandomando,

Thank you very much for submitting your manuscript "Escherichia coli ST131 clones harbouring AggR and AAF/V fimbriae causing bacteremia in Mozambican children: emergence of new variant of fimH27subclone" for consideration at PLOS Neglected Tropical Diseases. As with all papers reviewed by the journal, your manuscript was reviewed by members of the editorial board and by several independent reviewers. In light of the reviews (below this email), we would like to invite the resubmission of a significantly-revised version that takes into account the reviewers' comments. 

His manuscript is interesting and novel, but before being accepted it needs to be improved taking into account the most important comments made by the three reviewers. 

In addition, you should compare your results with those found by Roer et al in Denmark.

WGS-based surveillance of third-generation cephalosporin-resistant Escherichia coli from bloodstream infections in Denmark.

Roer L, Hansen F, Thomsen MCF, Knudsen JD, Hansen DS, Wang M, Samulioniené J, Justesen US, Røder BL, Schumacher H, Østergaard C, Andersen LP, Dzajic E, Søndergaard TS, Stegger M, Hammerum AM, Hasman H.

J Antimicrob Chemother. 2017 Jul 1;72(7):1922-1929. doi: 10.1093/jac/dkx092.

We cannot make any decision about publication until we have seen the revised manuscript and your response to the reviewers' comments. Your revised manuscript is also likely to be sent to reviewers for further evaluation.

Sincerely,

Jorge Blanco, Ph. D.

Guest Editor

Ana LTO Nascimento

Deputy Editor

His manuscript is interesting and novel, but before being accepted it needs to be improved taking into account the most important comments made by the three reviewers. 

In addition, you should compare your results with those found by Roer et al in Denmark.

WGS-based surveillance of third-generation cephalosporin-resistant Escherichia coli from bloodstream infections in Denmark.

Roer L, Hansen F, Thomsen MCF, Knudsen JD, Hansen DS, Wang M, Samulioniené J, Justesen US, Røder BL, Schumacher H, Østergaard C, Andersen LP, Dzajic E, Søndergaard TS, Stegger M, Hammerum AM, Hasman H.

J Antimicrob Chemother. 2017 Jul 1;72(7):1922-1929. doi: 10.1093/jac/dkx092.

Reviewer's Responses to Questions

**Key Review Criteria Required for Acceptance?**

**Methods**

-Are the objectives of the study clearly articulated with a clear testable hypothesis stated?

-Is the study design appropriate to address the stated objectives?

-Is the population clearly described and appropriate for the hypothesis being tested?

-Is the sample size sufficient to ensure adequate power to address the hypothesis being tested?

-Were correct statistical analysis used to support conclusions?

-Are there concerns about ethical or regulatory requirements being met?

Reviewer #1: (No Response)

Reviewer #2: Objectives and methods are appropiated.

Reviewer #3: Study design partially appropriate

**Results**

-Does the analysis presented match the analysis plan?

-Are the results clearly and completely presented?

-Are the figures (Tables, Images) of sufficient quality for clarity?

Reviewer #1: (No Response)

Reviewer #2: Results are described in a logical and straightforward manner.

Reviewer #3: results section needs clarification

**Conclusions**

-Are the conclusions supported by the data presented?

-Are the limitations of analysis clearly described?

-Do the authors discuss how these data can be helpful to advance our understanding of the topic under study?

-Is public health relevance addressed?

Reviewer #1: (No Response)

Reviewer #2: The discussionand conclusions provides a balanced assessment of the author’s findings.

Reviewer #3: only partially

**Editorial and Data Presentation Modifications?**

Reviewer #1: (No Response)

Reviewer #2: (No Response)

Reviewer #3: (No Response)

**Summary and General Comments**

Reviewer #1: This study examines E. coli ST131 causing bacteremia in Mozambican children that finds an associated between AggR and AAF/V fimbriae and Clade B isolates, predominating unexpectedly over Clade C. This counters the global trend, and so is of interest, as well as the region: there are really few samples from this geographic region, so that aspect alone is worth noting. Generally the study is good, though I have comments below on the nomenclature used, the stats in Table 1. A phylogeny would definitely enhance it to identify the fimH27 source. The major consideration is the genome assembly methods with the CLC workbench: no detail on the methods used is given, which undermines confidence in these results. More detail is below.

Comments:

- use the conventional Clade A/B/C nomenclature as well as of fimH41 / fimH22 / fimH30 - H27 is usually associated with subclade B0 (see Ludden et al 2019 https://doi.org/10.1101/814731, Ben Zakour et al 2016, Kallonen et al 2017, Stoesser et al 2016) or more likely just Clade B (Decano et al 2019 https://www.nature.com/articles/s41598-019-54004-5). The O sertoypes seem quite surprising in the context of previous work for ST131.

- Just FYI I believe the only other ST131 known to be from the Africa region are SRR1186472 and SRR1191664 (SRA accessions) from 2009 from Tanzania (many other related fimH27 samples do not have geographic information). Many others from outside Africa are available.

- Table 1 - are the 42 (approx) statistical tests applied and associated p values corrected for multiple testing? Some of the numbers in this table dom't add up - eg for genes aggR and aap the % of EAEC out of 44 is the same (41 and 41) but the % values are not. Later, no numbers are given for fimH and partially for agg4A. Please double-check. In addition, it would be easier for the reader IMHO if you split the table into two parts denoting the virulence factors associated with EAEC and those with the others: it is not easy to examine for ones in the latter category.

- This paper and inference of the ancestry of these isolates would be enhanced by a phylogeny of the core genome SNPs. Eg use Gubbins/ClonalFrameML to exclude recombinant tracts, then RAxML to construct the tree visualised with iTOL or FigTree. This would answer if they are the same outbreak, independent introductions from

- For the genome sequencing, much more detail is needed:

* How many reads were generated per sample? What were the mean and SD of their lengths, and likewise of their insert sizes?

* What QC (eg Fastp, etc) was done on the reads? Did you do base correction? (eg with BayesHammer/SPAdes/Pilon) (see Alikhan et al 2018 for an equivalent alternative approach)

* What were the assembly N50s, distribution of contig size lengths and how do you know the assembly process worked well? It is quite possible that some non-detected genes are a consequence of poorly assembled regions. Use Quast to look at the N50s, numbers of predicted genes/ORFs and numbers of contigs with mis-assemblies. Example standards as per Alikhan et al 2018 are assembly lengths of 3.7 to 6.4 Mb with less than 800 contigs and under 5% low-quality sites.

* How did you measure the assembly success say compared to the best available assembler Unicycler? Can you show that CLC generates assemblies not markedly inferior to this?

* What E. coli reference genome was used during the assembly process for scaffolding? (eg SSPACE, Mauve, etc)

* Did you annotate the genomes? (eg Prokka)

Reviewer #2: General comments:

This is an interesting, novel and clearly written manuscript. Results are described in a logical and straightforward manner. The discussion provides a balanced assessment of the author’s findings. In my view, one of the limitations of the study is the lack of information about the HIV infection in the population studied, considering that is performed in one of the regions with highest sero-prevalence of HIV in the world. HIV infections and immunosuppressive status generated by this infection could be a risk factor (maybe the more important) to develop an infection by these E. coli strains in the population studied.

Although this paper is focused on molecular epidemiology and virulence, a brief paragraph describing a bit more about antimicrobial resistance of the isolates studied and molecular mechanisms of resistance will bring even more interest to the manuscript. Authors, for instance says “our strains were susceptible to third generation cephalosporines and fluoroquinolones.” But, what about other antimicrobial families? Did any strain meet the MDR criteria? Line 332 says “WGS also identified genes conferring resistance towards three or more groups of antibiotics”, however this information does not indicate that they are MDR isolates.

CART analysis is interesting however it doesn´t take into account important variables of poor outcome such as antimicrobial resistance, adequacy of empiric and targeted treatments, HIV… This could be briefly discussed.

Could the authors classify the ST1313 isolates studied according to virotype classification (A to F)?

Specific comments:

Line 75, change symptoms by diseases.

Line 334: Change beta-lactamase by beta-lactams. Extend information about the antimicrobial encoding-genes found.

Table 3: Detail how the incidence rate was calculated.

Figure 2: Images have poor resolution

Line 420-421: Change “extended spectrum of β-lactamase” by broad spectrum β-lactams and “cephalosporines” by cephalosporins.

Reviewer #3: The paper of Mandomando and colleagues address the emergence of a new subclade of ST131 in a population of African bacteriemic children. Moreover, these isolates belonging to FimH27 contains both ExPEC and EAEC genes suggesting that this subclone may serve as a ‘melting pot’ for pathogroup conversion between EAEC and ExPEC. The paper is correctly written and may be of interest for readers however, I have some difficulties with some points. First, it is not clear what was done using PCR or serotyping and was done with WGS data analysis. Detection of pathotypes, virulence factors, and serotyping may be determined by in silico analysis. Secondly, among the 325 isolates collected for analysis, how were selected the 44 isolates for WGS? I understand that only EAEC isolates (identified with PCR methods) were sequenced. Is it true? How the authors justified this selection? Similarly, in the results section, the authors state that 66 isolates were serotyped, How the authors justified this selection?

Globally the collection analysis is not clear and the mode of isolate selection for further analysis should be more precisely described.

From my point of view, despite the novelty of the findings, the manuscript should be totally rewritten , I suggest the authors focused on description of emerging fimH27 ST131 subclade by comparing their own WGS data with ST131 WGS data available in NCBI

PLOS authors have the option to publish the peer review history of their article (what does this mean?). If published, this will include your full peer review and any attached files.

Reviewer #1: No

Reviewer #2: No

Reviewer #3: No
---

## [Editor Report · Decision Letter 1]

7 Apr 2020

Dear Dr. Mandomando,

We are pleased to inform you that your manuscript 'Escherichia coli ST131 clones harbouring AggR and AAF/V fimbriae causing bacteremia in Mozambican children: emergence of new variant of fimH27subclone' has been provisionally accepted for publication in PLOS Neglected Tropical Diseases.

Best regards,

Jorge Blanco, Ph. D.

Guest Editor

Ana LTO Nascimento

Deputy Editor

---

## [Editor Report · Acceptance letter]

23 Apr 2020

Dear Dr. Mandomando,

We are delighted to inform you that your manuscript, "Escherichia coli ST131 clones harbouring AggR and AAF/V fimbriae causing bacteremia in Mozambican children: emergence of new variant of fimH27subclone," has been formally accepted for publication in PLOS Neglected Tropical Diseases.

Best regards,

Serap Aksoy

Editor-in-Chief

Shaden Kamhawi

Editor-in-Chief
